# Risk factors for disease severity and increased medical resource utilization in respiratory syncytial virus (+) hospitalized children: A descriptive study conducted in four Belgian hospitals

Marijke Proesmans[1], Annabel Rector[2], Els Keyaerts[2], Yannick Vandendijck[3], Francois Vermeulen[1], Kate Sauer[4], Marijke Reynders[5], Ann Verschelde[6], Wim Laffut[7], Kristien Garmyn[8], Roman Fleischhackl[9], Jacques Bollekens[3], Gabriela Ispas[3]*

1 Department of Pediatrics, University Hospitals Leuven, Leuven, Belgium, 2 KU Leuven Department of Microbiology, Immunology and Transplantation, Rega Institute, Laboratory of Clinical and Epidemiological Virology, Leuven, Belgium, 3 Janssen Pharmaceutica, Beerse, Belgium, 4 Department of Pediatrics, AZ Sint-Jan Brugge—Oostende, Campus Brugge, Brugge, Belgium, 5 Department of Microbiology, AZ Sint-Jan Brugge—Oostende, Campus Brugge, Brugge, Belgium, 6 Department of Pediatrics, AZ Sint-Jan Brugge–Oostende, Campus Henri Serruys, Oostende, Belgium, 7 Department of Microbiology, Heilig-Hartziekenhuis, Lier, Belgium, 8 Department of Pediatrics, Heilig-Hartziekenhuis, Lier, Belgium, 9 Janssen-Cilag Austria, Vienna, Austria

* gispas@its.jnj.com

**Data Availability Statement:** The data sharing policy of Janssen Pharmaceutical Companies of

## Abstract

### Background

We aimed to provide regional data on clinical symptoms, medical resource utilization (MRU), and risk factors for increased MRU in hospitalized respiratory syncytial virus (RSV)-infected Belgian pediatric population.

### Methods

This prospective, multicenter study enrolled RSV (+) hospitalized children (aged ≤5y) during the 2013–2015 RSV seasons. RSV was diagnosed within 24h of hospitalization. Disease severity of RSV (+) patients was assessed until discharge or up to maximum six days using a Physical Examination Score (PES) and a derived score based on ability to feed, dyspnea and respiratory effort (PES3). MRU (concomitant medications, length of hospitalization [LOH], and oxygen supplementation) was evaluated. Kaplan-Meier survival analysis was performed to compare MRU by age and presence of risk factors for severe disease. Association between baseline covariates and MRU was analyzed using Cox regression models.

### Results

In total, 75 children were included, Median (range) age was 4 (0–41) months, risk factors were present in 18.7%, and early hospitalization (≤3 days of symptom onset) was observed in 57.3% of patients. Cough (100%), feeding problems (82.2%), nasal discharge (87.8%),

Johnson & Johnson is available at https://www.janssen.com/clinical-trials/transparency. As noted on this site, requests for access to the study data can be submitted through Yale Open Data Access (YODA) Project site at http://yoda.yale.edu.

**Funding:** The study was funded by Janssen Pharmaceutica NV. The funders had no role in study design, data collection and analysis, decision to publish, or preparation of the manuscript.

**Competing interests:** I have read the journal's policy and the authors of this manuscript have the following competing interests: YV, RF, JB, and GI are employees of Janssen Pharmaceutica NV and may own stock in Johnson & Johnson. MP, AR, EK, FV, KS, MR, AV, WL, and KG have no conflict of interest.

and rales and rhonchi (82.2%) were frequently observed. Median (range) LOH and oxygen supplementation was 5 (2–7) and 3 (1–7) days. Oxygen supplementation, bronchodilators, and antibiotics were administered to 58.7%, 64.0%, and 41.3% of the patients, respectively. Age <3 months and baseline total PES3 score were associated with probability and the duration of receiving oxygen supplementation. LOH was not associated with any covariate.

## Conclusion

RSV is associated with high disease burden and MRU in hospitalized children. Oxygen supplementation but not length of hospitalization was associated with very young age and the PES3 score. These results warrant further assessment of the PES3 score as a predictor for the probability of receiving and length of oxygen supplementation in RSV hospitalized children.

## Registration

NCT02133092

## Introduction

RSV is a common cause of respiratory tract infection in infants and young children [1], and was estimated to be responsible yearly for at least 33.1 million episodes of acute lower respiratory tract infection worldwide in 2015, resulting in the death of 94,600 to 149,400 children aged <5 years [2]. It has been documented that the clinical presentation of RSV infection in children differs according to age and may be influenced by the differences in their immune reaction to RSV [3]. In a metareview, it was shown that the annual RSV hospitalization rates decreased with increasing age and varied by a factor of 2–3. Risks factors associated with RSV related medical resources utilization included, male sex; age <6 months; birth during the first half of the RSV season; crowding/siblings; and day-care exposure (high strength of evidence) [4].

RSV activity is influenced by meteorological variables such as temperature, humidity, air pressure [5], and geographical latitude [6]. The European Influenza Surveillance Network reported the median RSV peak to be in late January and early February in the Northern hemisphere [6]. However, the pattern is different in Belgium, as RSV infection peaks in young children usually in November or December [1] and around six weeks later in the elderly [7].

Similar to findings from other countries, Cattoir et al. [1] reported the proportion of reverse transcription polymerase chain reaction (RT-PCR) positive samples for RSV to be greater in younger children aged <4 years than those aged >4 years (23% vs. 4%) in Belgium, with infants aged <6 months being affected more than other age groups. A recent systematic review also reported the highest number of RSV-related hospitalizations (>1.4 million) globally among infants aged <6 months [2], and the average annual hospitalization rate for RSV infection in Danish infants aged <6 months was 45.9/1000 [8]. Similar studies in the United Kingdom have shown that infants aged <6 months account for >40% of all RSV-related hospitalizations [9, 10].

Although Belgium is affected by RSV every year [1, 7], local data regarding hospitalization, medical resource utilization (MRU), as well as RSV disease progression for children aged ≤5 years are unavailable. This study aimed to provide evidence on factors associated with length

of hospitalization and request for oxygen supplementation in RSV (+) hospitalized Belgian children. Both demographic characteristics (age, presence of underlying risks) and disease severity were considered as potential drivers for MRU.

## Methods

### Study design

This exploratory, prospective, multicenter study (NCT02133092) enrolled patients from clinical pediatric wards of four Belgian hospitals (one tertiary academic center and three regional hospitals: UZ Leuven, AZ Sint-Jan Brugge–Oostende campus Brugge, AZ Sint-Jan Brugge—Oostende campus Henri Serruys, and Heilig Hart Ziekenhuis Lier) during the 2013–2014 and 2014–2015 RSV epidemic seasons (Study initiated: 17 December 2013 and Study completed: 21 January 2015). Patients were diagnosed as RSV (+) based on a quantitative RT-PCR (qRT-PCR) homemade/in-house test, or a Sofia RSV fluorescent immunoassay (SOFIA® RSV tests, Quidel), whichever was available first, within 24h after hospitalization. Children with a positive RSV test were monitored daily until discharge or for a maximum of six days (Day 2 to Day 7 of hospitalization).

RSV disease severity, peripheral oxygen saturation (SpO2), respiratory rate and heart rate were monitored.

Approval for the study was obtained from the Commissie Medische Ethiek of the Universitaire Ziekenhuizen Leuven (study reference number—S55858).

### Study population

Pediatric patients aged ≤5 years with a diagnosis of lower respiratory tract infection (LRTI) as per the hospital's standard-of-care (SOC) were enrolled if: (a) they were hospitalized for ≤24 hours duration, (b) the legal representative had provided an ethics committee-approved written informed consent and (c) the patient had a confirmed RSV infection.

### Assessments

**Demographics and clinical characteristics.** Age, weight, gender, current breastfeeding status, day care attendance, duration of symptoms, and underlying risk factors were assessed at enrolment. Duration of symptoms prior to enrolment was collected by asking the caregiver the following question "When did the first acute respiratory symptoms appear?". The following signs and symptoms were considered as first manifestations of the disease: apnea and feeding problems/ reduction of food intake (for neonates) or being irritable, rhinorrhea, nasal congestion, pharyngitis, cough, ear pain (for older children). Underlying conditions i.e., risk factors reported by the parent/guardian were documented, such as asthma/atopy, previous or recurrent wheezing episode, congenital lung disease, congenital heart disease, immunodeficiency, and bronchopulmonary dysplasia and other risks. Prematurity was documented and defined as ≤37 weeks of gestation, however, no information was available regarding the prematurity sub-category.

**Disease severity monitoring.** RSV disease severity was monitored by recording clinical signs and symptoms every morning. The signs and symptoms were recorded prior to nasal specimen collection. SpO2 was assessed on room air.

The Physical Examination Score (PES) was used to assess the disease progression over time, as reported by clinician. The PES was developed based on the Emboriadou-Garofalo Bronchiolitis Score [11, 12], Pedianet Bronchiolitis Score [13], and DeVincenzo et al.'s method [14]. Items included in the PES were: ability to feed; otitis; cough; nasal discharge; dyspnea;

respiratory efforts; and lung sounds (rales, rhonchi, and wheezing). Each parameter was scored on a four-point severity scale (0 = no symptoms to 3 = severe symptoms) for a total score ranging from 0 to 24 (S1 Table in S1 File). A secondary score termed the PES3 score, was also used. The PES3 score was based on main signs/symptoms with major clinical relevance, such as ability to feed, dyspnea and respiratory effort. The sum score of ability to feed, dyspnea and respiratory effort was used for further assessments.

**Medical resource utilization.** Medical supportive care (mechanical ventilation and supplementary oxygen), concomitant medication, length of hospitalization (LOH, in days), and physician visits for acute respiratory infection (ARI) before hospitalization were documented.

## Statistical analysis

Demographic and clinical characteristics were tabulated and analyzed descriptively and graphically. Data were descriptively analyzed per age groups (0–<3 months, 3–<6 months, 6–<12 months, and 12–48 months), duration of symptoms before hospitalization (early presentation: ≤3d and late presentation: >3d–≤5d), and presence of underlying risk factors.

Several analyses were conducted to examine the risk factors associated with LOH, need for and duration of oxygen supplementation. Kaplan-Meier survival analysis was performed to compare LOH and length of oxygen supplementation across age groups and underlying risk factors (log-rank test). Univariate and multivariate logistic regression models were applied to investigate whether covariates impact the dependent variables LOH (categorized as LOH ≤4 days and LOH >4 days) and the need for supplemental oxygen (subject received supplemental oxygen during hospitalization: Yes/No). Univariate and multivariate Cox proportional hazard regression models [15] were implemented on the LOH (in days) and length of oxygen supplementation (in days).

The following, at baseline available, independent variables (covariates) were considered: demographic variables [age, sex], comorbidities [underlying risk], duration of symptoms before hospitalization, signs and symptoms severity [individual PES3 items and total-PES3 score on day 1]. In addition, reception of oxygen supplementation on day 1 (Yes/No) was included as a covariate for the modeling of LOH. Three options were explored to incorporate the PES score on day 1: (1) the total PES score by summing the 8 individual components; (2) scores on the individual components of ability to feed, dyspnea and respiratory effort; and (3) the PES3 score. The results of options (2) and (3) are presented here.

Statistical analyses were performed using SAS software package (version 9.2 for Windows, SAS Institute Inc., Cary, NC, USA) and R software package (R version 3.6.0 for Windows) [16, 17] Baseline characteristics were compared using Wilcoxon rank sum test or chi-square test where applicable. P-values without multiple comparison corrections were reported. All significance tests were 2-sided with a 5% significance level, where applicable.

## Results

### Baseline characteristics

Seventy-five patients were included in the study. Baseline characteristics were reported for all patients (Table 1) as well as after classifying based on age, symptom duration, and underlying risk (S3 Table in S1 File). Overall, the median (range) age was 4 (0–41) months with more males than females (n = 41/75 [54.7%] vs. n = 34/75 [45.3%]). Majority of patients were infected by RSV-A (n = 68/75 [90.7%]). Underlying risk factors were present in 18.7% of the patients (n = 14/75), with previous or recurrent wheezing (n = 5/75) and congenital heart disease (CHD; n = 4/75) being the most common underlying risk factors. Premature birth (≤37 weeks of gestation) was reported in 20.0% (n = 15/75) of the patients, with a

**Table 1. Baseline characteristics and medical resource utilization.**

| Parameter | All (n = 75) |
|---|---|
| **Age (months, median [range])[a]** | 4.0 (0–41) |
| **Age (months, n [%])[a]** | |
| 0 - <3 | 28 (37.8) |
| 3 - <6 | 15 (20.3) |
| 6 - <12 | 13 (17.6) |
| 12–48 | 18 (24.3) |
| **Gender (n [%])** | |
| Female | 34 (45.3) |
| Male | 41 (54.7) |
| **RSV subtype (n [%])** | |
| A | 68 (90.7) |
| B | 7 (9.3) |
| **Weight at birth (kg, median [range])** | 3.34 (1.00–4.37) |
| **Baseline weight (kg, median [range])** | 6.91 (2.68–17.70) |
| **Day care attendance (n [%])[a]** | |
| Yes | 36 (48.6) |
| No | 38 (51.4) |
| **Currently breastfed (n [%])** | |
| Yes | 27 (36.0) |
| No | 48 (64.0) |
| **Underlying risk[b] (n [%])** | |
| Yes | 14 (18.7) |
| No | 61 (81.3) |
| **Premature birth (n [%])[f]** | |
| Yes | 15 (20.0) |
| No | 60 (80.0) |
| **Symptom length (days, median [range])** | 3.0 (1–5) |
| **Symptom length (days, n [%])** | |
| ≤3 | 43 (57.3) |
| >3 | 32 (42.7) |
| **Co-medication during study (n [%])** | |
| Antibiotics | 31 (41.3) |
| Bronchodilators | 48 (64.0) |
| Corticosteroids | 6 (8.0) |
| Others[c] | 4 (5.3) |
| **Length of hospital stay (days, median [range])[d]** | 5.0 (2–7) |
| **Oxygen supplementation (n [%])** | |
| Yes | 44 (58.7) |
| No | 31 (41.3) |
| **Length of oxygen supplementation (days, median [range])[e]** | 3.0 (1–7) |
| **Visited family doctor for ARI before hospitalization (n [%])** | |
| Yes | 39 (52.0) |
| No | 36 (48.0) |

[a] The sample size for the evaluation was 74 since data was missing for 1 patient.

[b] Underlying risk includes atopy (n = 2), previous or recurrent wheezing (n = 5), CHD (n = 4), immunodeficiency (n = 1) and others (n = 4).

[c] One patient received Synagis (older age group, likely immunocompromised and presented early [≤3d]).

[d] The sample size for this evaluation was 61.

[e] The sample size for this evaluation was 44.

[f] Prematurity defined as (≤37 weeks of gestation)

**Abbreviations:** ARI–Acute Respiratory Infection, CHD–Congenital Heart Disease, RSV–Respiratory Syncytial Virus, SD–Standard Deviation

median (range) age of 7 (0–28) months at enrolment. The premature group was elder at the time of hospital admission than the patients without reported prematurity (median [range] age of 3 [0–41] months). Median (range) symptom duration at enrolment was 3.0 (1–5) days, with symptoms being reported for ≤3 days for most patients (n = 43/75 [57.3%]) before being hospitalized.

Compared with the patients with early presentation (≤3 days from symptom onset), patients with later presentation (>3 days from symptoms onset) showed a trend for older age (median age 5.5 vs. 2.5 months), less current breastfeeding (28.1% vs 41.9%), and had more premature births (25% vs 16.3%). However, none of these differences were statistically significant. Fewer patients had underlying risk factors in the late intercept group (n = 3/32 [9.4%]) than the early intercept group (n = 11/43 [25.6%], p = 0.138) (S2 Table in S1 File).

Patients with underlying risk factors (n = 14) were older (median [range] age 8.0 [1–32] vs. 3.0 [0–41] months, p = 0.056), were breastfed less (n = 4/14 [28.6%] vs. n = 23/61 [37.7%], p = 0.739), and had a shorter symptom duration at hospitalization (median [range] 2.5 [1–5] vs. 3.0 [1–5] days, p = 0.078) than patients without underlying risk factors (n = 61) (S3 Table in S1 File).

## Medical resource utilization

**Concomitant medications.** Bronchodilators (n = 48/75 [64.0%]) and antibiotics (n = 31/75 [41.3%]) were the most common medications prescribed overall (Table 1) and in all the subgroups (see S3 Table in S1 File). Patients with underlying risk factors received more antibiotics than patients without underlying risk factors (n = 10/14 [71.4%] vs. n = 21/61 [34.4%], p = 0.025; S4 Table in S1 File).

**Hospitalization.** Overall, the median (range) LOH was 5.0 (2–7) days (Table 1), with small variations between a minimum of 4.0 (2–6) days and a maximum of 6.0 (2–7) days between different groups (S3 Table in S1 File). Overall, 8 patients (10.8%) were hospitalized for >7 days; 5 of these 8 patients were from the youngest age group (<3 months).

**Oxygen supplementation.** Overall, oxygen supplementation was given to 44/75 (58.7%) of the patients for a median (range) duration of 3.0 (1–7) days (Table 1). The need for oxygen supplementation was highest in the 0–<3 months group (n = 23/28 [82.1%]) for a median (range) duration of 4.0 (1–7) days. Proportion of patients who received oxygen supplementation and the duration of supplementation was similar in patients with symptoms for ≤3 days and >3 days. Underlying risk factors did not result in greater usage of oxygen supplementation (S3 Table in S1 File).

**Prior physician visit.** Half of the patients (n = 39/75 [52.0%]) visited their family doctor prior to hospitalization (Table 1). Patients aged 3–<6 months (n = 11/18 [61.1%], p = 0.046) and those with symptoms for >3 days (n = 22/32 [68.8%], p = 0.023), visited their physicians the most prior to being hospitalized (S3 Table in S1 File).

## Clinical disease kinetics

Patients were mostly afflicted with cough (100%–93. 7%), feeding problems (82.2%–37.5%), nasal discharge (93.2%–81.2%), and rales or rhonchi (84.9%–40.0%) throughout the assessment period (Fig 1). The mean (SE) PES item score on each assessment day as per underlying risk, age, and symptom onset, respectively are illustrated in Fig 2 and S4 and S5 Figs in S1 File. Higher scores (i.e., worse condition) in the ability to feed item were noted for the 3–6 months group (S5 Fig in S1 File). Scores for wheezing, rales and ronchi, dyspnea, respiratory effort, and otitis were higher for patients with underlying risk factors (Fig 2).

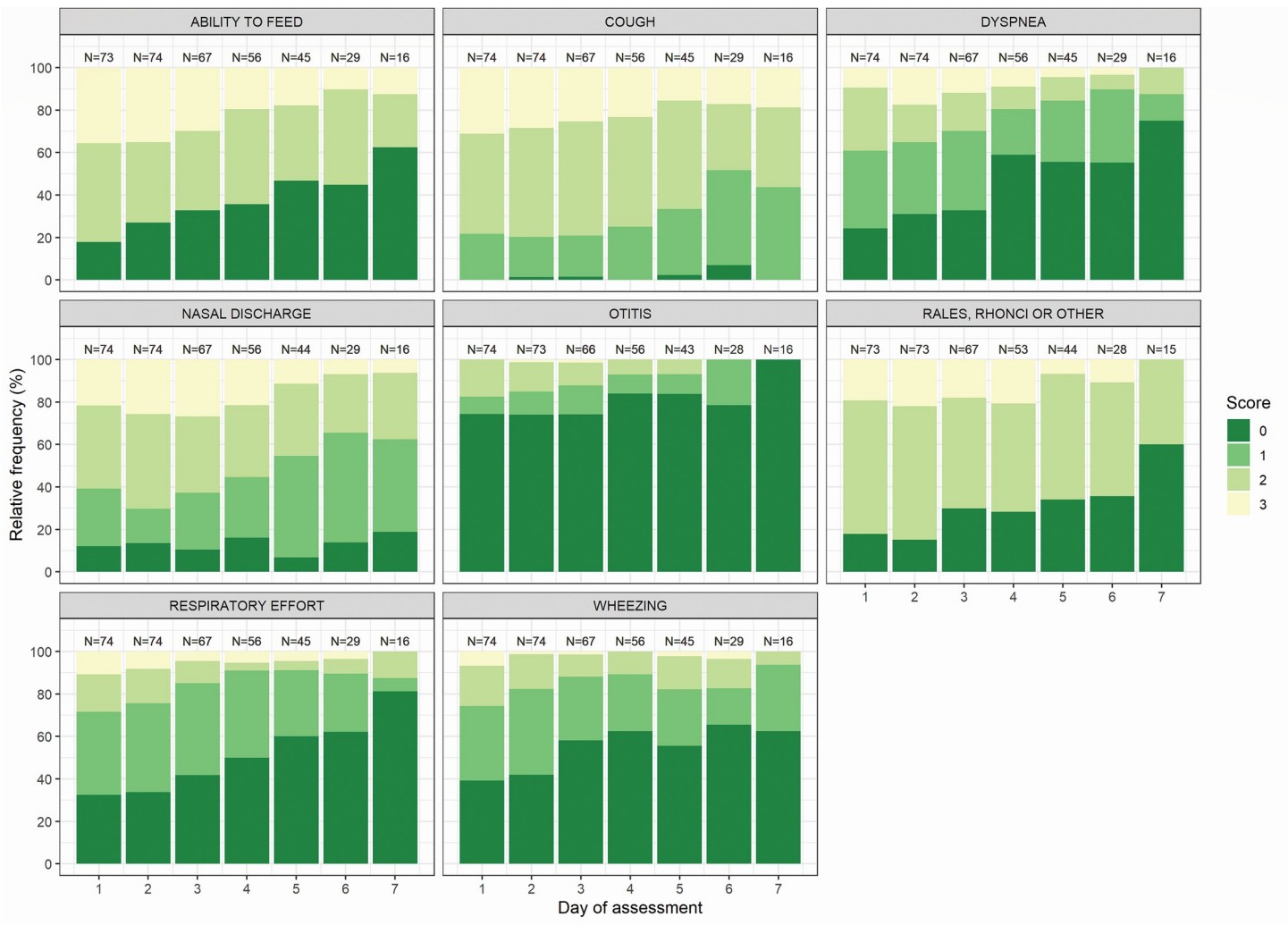

**Fig 1. PES item frequency over time as per item severity.** Relative frequency of each PES item is expressed as per item severity in all the patients. Item severity is scored from 0 to 3. Patients who were discharged were not considered in the following days' calculation which could bias the improvement observed over time. **Abbreviations:** PES–Physical Examination Scoring.

### Factors affecting length of hospitalization

The LOH was not statistically different between the age groups (log-rank test, p = 0.320) and by presence of underlying risks (log-rank test, p = 0.884; S6 Fig in S1 File). For the logistic regression assessment, LOH was categorized into: (1) patients with a LOH ≤4d (n = 27), and (2) LOH >4d (n = 45). When baseline covariates were assessed, none of the covariates were observed to significantly influence LOH in both the univariate and multivariate regression models (S7 Table in S1 File). Cox regression analysis showed similar results (S8 Table in S1 File).

### Factors affecting oxygen supplementation

Patients aged 0–<3 months received oxygen supplementation for a longer duration than other age groups (log-rank test, p = 0.0016). Duration of oxygen supplementation was not statistically different between patients with and without underlying risk factors (log-rank test, p = 0.084; Fig 3).

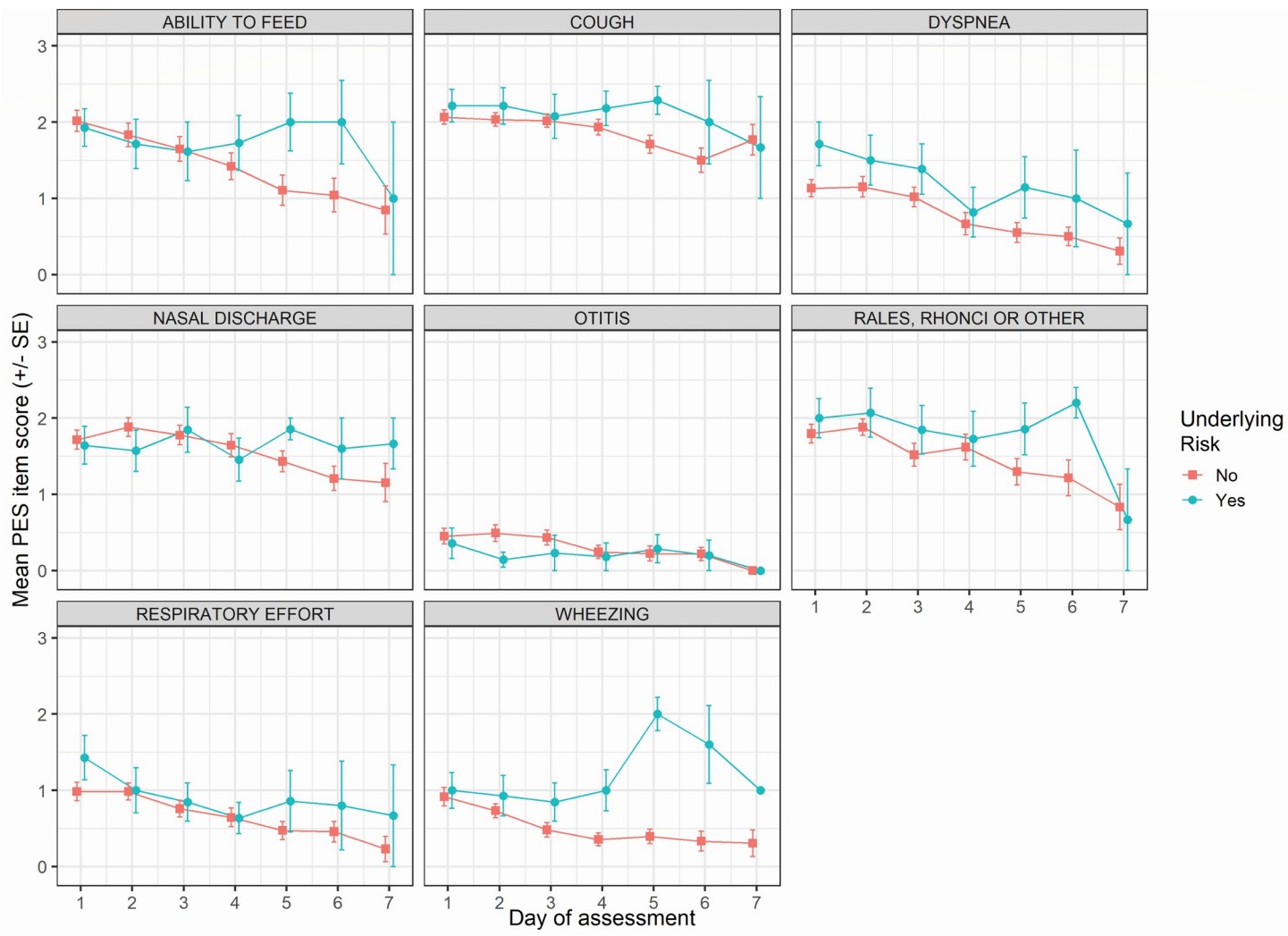

**Fig 2. PES item score over time as per underlying risk.** Mean ± SE PES item score is represented at each day of assessment for patients classified based on presence of absence of underlying risk. Patients who were discharged were not considered in the following days' calculation which could bias the improvement observed over time. **Abbreviations:** PES–Physical Examination Scoring, SE–Standard Error.

Patients were categorized based on whether they received supplemental oxygen (n = 29) or not (n = 43) during hospitalization. Age significantly affected the probability of receiving supplemental oxygen (p = 0.005; S9 Table in S1 File). Patients in the youngest age category had a higher probability of receiving supplemental oxygen. Furthermore, there was a significant effect of the PES3 score at Day 1 on the probability of receiving supplemental oxygen, with an estimated odds ratio for a 1-unit increase in PES3 score of 1.63 (95% CI 1.19–2.37).

Cox regression analysis showed similar results with a significant effect of age (p = 0.001) and Day 1 PES3 score (p = 0.029) on the length of receiving oxygen supplementation. Patients aged 3–6 months were at a lower risk of receiving oxygen supplementation during hospitalization when compared with patients aged 0–<3 months (hazard ratio [HR] 3.76 (95% CI 1.75–8.11) (Table 2). This corresponds to a 79% chance (HR/[1-HR]) [18] that patients aged 3–6 months have a shorter oxygen supplementation period compared to patients aged 0–<3 months. The HR of the PES3 score was significantly lower than 1 (0.86 [95% CI 0.75–0.99];

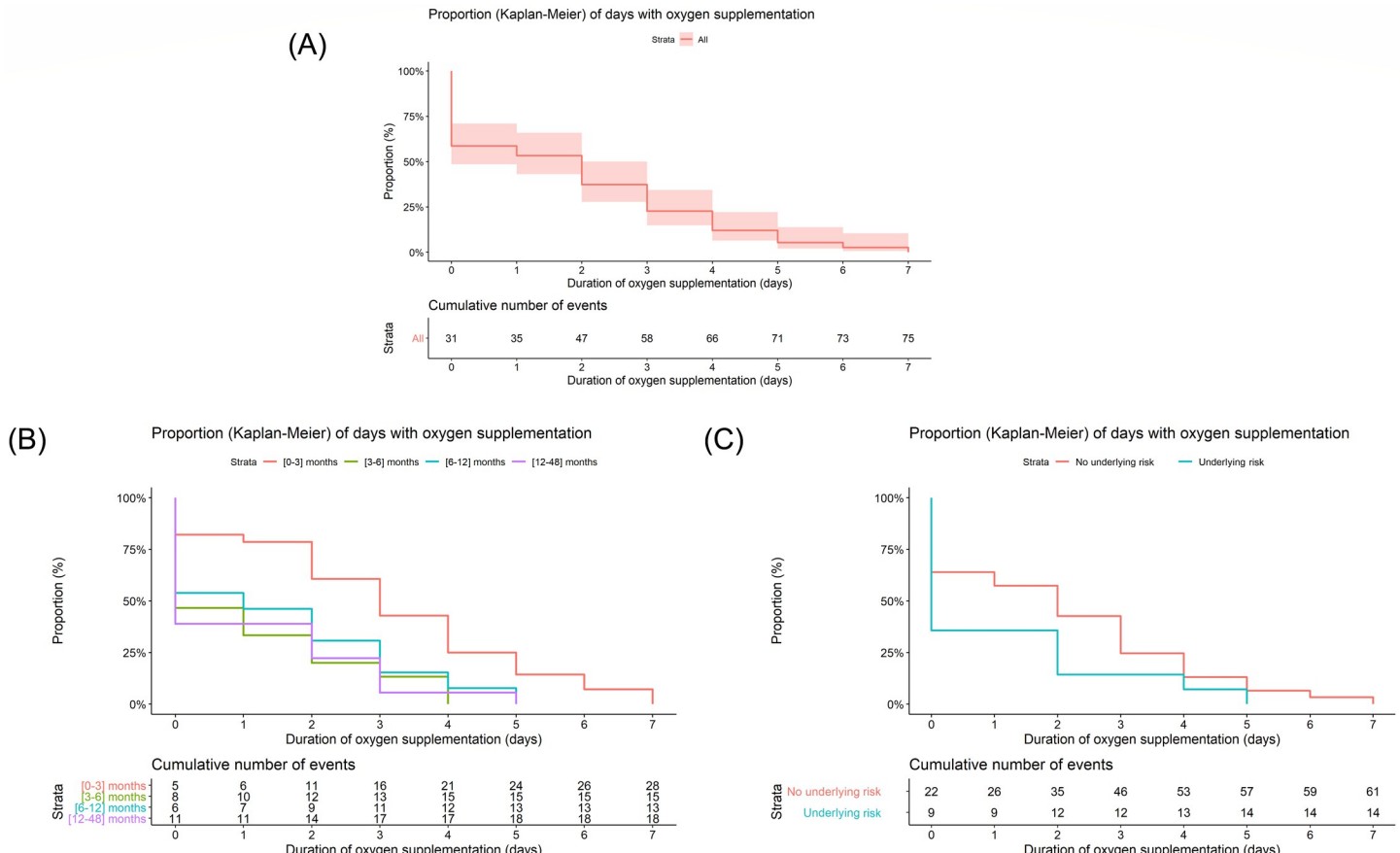

**Fig 3. Length of oxygen supplementation by age and underlying risk.** Kaplan-Meier curves represent the length of oxygen supplementation for (A) overall patients, (B) patients classified based on age, and (C) patients classified based on underlying risk.

p = 0.029), indicating that a higher PES3 score on day 1 was associated with a longer duration of oxygen supplementation.

## Discussion

This study describes the disease progression, medical resources utilization and predictors for oxygen supplementation and duration of hospitalization. in RSV (+) children hospitalized with a LRTI. There were no transfers to ICU or mechanical ventilation (although one of the four sites was a tertiary hospital), indicating that the population studied is representative for moderate severity bronchiolitis hospitalizations.

Disease severity was characterized based on clinical evaluation of signs and symptoms covering the full spectrum of LRTIs, as suggested by Karron and Zar [19]. A greater severity of otitis and nasal discharge was observed among older children (>6 months), and greater respiratory effort among younger children (<6 months) over time. We observed a greater severity of dyspnea and greater respiratory effort in neonates (<3 months), whereas Ogra [20] reported that reduction in food intake and apnea were the most severe findings in this age group. The median (range) LOH for the overall patient population was 5.0 (2–7) days. This duration was similar compared with studies in children in France [21], Spain [22], and Japan [23] and longer compared with studies from England [24], the United States [25], and New Zealand [26].

**Table 2. Cox proportional hazard regression analysis for number of days a patient received oxygen supplementation.**

| Parameter | Multivariate analysis | |
|---|---|---|
| | **HR (95% CI)** | **p value[a]** |
| **Age** | | |
| 0–<3 months | - | 0.001 |
| 3–6 months | 3.76 (1.75–8.11) | |
| 6–<12 months | 2.94 (1.37–6.29) | |
| 12–<48 months | 3.81 (1.78–8.13) | |
| **Gender** | | |
| Female | - | 0.976 |
| Male | 1.01 (0.62–1.65) | |
| **Underlying risk** | | |
| No | - | 0.202 |
| Yes | 1.61 (0.79–3.31) | |
| **Length of symptoms at intercept** | | |
| ≤3 days | - | 0.389 |
| >3 days | 0.78 (0.44–1.37) | |
| **PES3-total score (3 items; 1-unit increase)** | 0.86 (0.75–0.99) | 0.029 |

N = 72, 2 patients were excluded as they did not have PES score on day 1 available. 1 patient was excluded due to missing age.

[a]p value was calculated by a likelihood ratio test.

A global chi-square test for the proportional hazard assumption showed no deviation (p = 0.447). A graphical review of possible time-dependent coefficients over time (plots of residuals for individual predictors) showed no deviation from the proportional hazard assumption.

**Abbreviations:** CI–Confidence Interval, HR–Hazard Ratio, PES–Physical Examination Scoring

Overall, 41.3% of the patients in the current study were prescribed antibiotics, which is similar to values reported by studies in Israel and the Netherlands (49%) [27]. Canada (60.5%) [28], and Germany (41.7%) [29]. Although patients with severe RSV infection receive antibiotics to treat a possible bacterial superinfection [30], doing so regardless of RSV severity may lead to adverse events and unnecessary financial burden [28] or antibiotic resistance. This finding highlights the overuse of antibiotics for treatment of viral related ARIs and the requirement for appropriate antibiotic stewardship programs [31–33]. We did not collect data on bacterial co-infection; however, we suspect overuse of antibiotics based on the facts that numerous studies have shown that the occurrence of a secondary or concurrent bacterial infection in hospitalized children with RSV lower respiratory tract disease. Nevertheless, frequency of RSV and bacterial co-infection in children is very low [34, 35]. The overuse of antibiotics for treatment of RSV disease has been documented before [27, 33].

The LOH is subjected to various factors (regional and site hospitalizations and discharge guideline, socio-economic reasons, etc.), partially independent of disease severity and therefore hampering the use of LOH as an endpoint for development of pharmaceutical interventions for ARI treatment [36, 37]. The current study did not find any demographic or clinical risk factor to significantly predict LOH. In contrast, in other studies, baseline characteristics such as male sex, lower weight, presence of congenital anomalies, etc. were associated with prolonged hospitalization [38–40]. Differences in study design, sample size, analysis methods and demographics could account for the conflicting data, i.e. younger children and ICU

hospitalization were included in the El Saleeby et al. study [41]. Compared with the current study (mean age 3 vs. 8 months and 25% ICU admission vs. no ICU admission).

We did not observe a relationship between the start of oxygen supplementation and LOH, whereas published evidence is conflicting. While Unger et al. [42] and Schroeder et al. [43] reported longer LOH in the presence of oxygen supplementation, a recent study reported that oxygen supplementation could have a variable influence on LOH depending on disease severity [44]. In the current study, young age (<3 months) but not disease severity were associated with greater probability of receiving oxygen as well as with length of oxygen supplementation, consistent with Stollar et al.'s results [45]. However, evidence regarding the association between severity assessment and oxygen supplementation is conflicting [46, 47] possibly due to use of different scoring systems. Our study shows an association between the PES3 score and probability of receiving oxygen and the length of oxygen supplementation. At our knowledge, this is the first time when the ability to feed, dyspnea and respiratory effort are combined in a composite score and predict the use of oxygen supplementation during hospitalization. Such a score, based on critical major signs/symptoms, would be easy to implement in clinical practice.

This study has limitations. Due to practical reasons, only a subset of patients admitted for RSV infections were enrolled in the study, therefore we cannot comment on the generalizability of the results. Most of the children are discharged with ongoing signs/symptoms that may drive additional medical resources utilization after hospital discharge. In this study, the MRUs were not captured after discharge, even though we consider that 7 days is sufficient time for capturing the utilization of medical resources [22, 48]. Since, only the current breastfeeding status was collected, the study was not able to evaluate the potential impact of prior breastfeeding on clinical burden and medical resources utilization during the hospitalization, In addition, the way different sites recruited participants could be different (some sites could have included mainly sicker patients that have a higher probability of receiving supplemental oxygen). We realized that the patients included may not be representative for all RSV related hospitalizations in the participating sites. Since a strict protocol on initiation and discontinuation of oxygen supplementation was not implemented, all participating hospitals followed their in-house protocol, which may vary from each other. The participating hospitals also did not have documented discharge criteria. This difference is a potential factor for bias introduced by site effects. Although not excluded in the protocol, no intensive care cases were included, hence most severe cases were not captured. No information was available on concomitant or secondary bacterial respiratory tract infections. However, the high use of antibiotics (41.3%) in our study may reflect lack of antibiotic stewardship and high suspicion of a concomitant bacterial infection, although bacterial infections are expected to occur with low frequency in RSV (+) hospitalized children. The RSV subtype was determined by first testing the sample for the predominant RSV subtype during the season. Further testing for the non-predominant RSV subtype was performed only when the samples were negative for the predominant subtype. Hence, patients with subtype A/B coinfections were not identified. Finally, the study results reflect MRUs related with the pediatric RSV hospitalization prior to the COVID19 pandemic, whereas the patient management pathway and associated burden during and post-pandemic may be different [49–53].

## Conclusion

This exploratory, prospective study provides evidence on disease burden and predictors for increased use of medical resources in hospitalized children. Very young age (<3 months old) and baseline PES3 total score were associated with the probability of receiving and the length

of oxygen supplementation, however, none of the factors analyzed were associated with the length of hospitalization. This study emphasizes that the probability and length of oxygen supplementation but not the length of hospitalization could be predicted in hospitalized children.

## Supporting information

**S1 File.**
(ZIP)

## Acknowledgments

The authors thank Dr. Marc Van Ranst and Dr. Lieselot Houspie for advising on the study conductance, and Leo J. Philip Tharappel (SIRO Clinpharm Pvt Ltd.), Aafrin Khan (SIRO Clinpharm Pvt Ltd) and Robert Achenbach (Janssen Global Services, LLC) for providing medical writing support and editorial assistance.

## Author Contributions

**Conceptualization:** Annabel Rector, Els Keyaerts, Jacques Bollekens, Gabriela Ispas.

**Data curation:** Marijke Proesmans, Annabel Rector, Els Keyaerts, Francois Vermeulen, Kate Sauer, Marijke Reynders, Ann Verschelde, Wim Laffut, Kristien Garmyn.

**Formal analysis:** Marijke Proesmans, Annabel Rector, Els Keyaerts, Yannick Vandendijck, Gabriela Ispas.

**Methodology:** Els Keyaerts, Gabriela Ispas.

**Supervision:** Roman Fleischhackl.

**Writing – original draft:** Annabel Rector, Els Keyaerts.

**Writing – review & editing:** Marijke Proesmans, Annabel Rector, Els Keyaerts, Francois Vermeulen, Kate Sauer, Marijke Reynders, Ann Verschelde, Wim Laffut, Kristien Garmyn, Roman Fleischhackl, Jacques Bollekens, Gabriela Ispas.

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
