## [Decision Letter · Decision Letter 0]

28 Sep 2021

PONE-D-21-25349Risk factors for disease severity and increased medical resource utilization in respiratory syncytial virus (+) hospitalized children: a descriptive study conducted in four Belgian hospitalsPLOS ONE

Dear Dr. Ispas,

Thank you for submitting your manuscript to PLOS ONE. After careful consideration, we feel that it has merit but does not fully meet PLOS ONE’s publication criteria as it currently stands. Therefore, we invite you to submit a revised version of the manuscript that addresses the points raised during the review process.

We look forward to receiving your revised manuscript.

Kind regards,

Brenda M. Morrow, PhD

Academic Editor

PLOS ONE

Journal Requirements:

2. Please note that PLOS does not permit references to 'results/data not shown.' Authors should provide the relevant data within the manuscript, the Supporting Information files, or in a public repository. If the data are not a core part of the research study being presented, we ask that authors remove any references to these data

Reviewers' comments:

Reviewer's Responses to Questions

**Comments to the Author**

1. Is the manuscript technically sound, and do the data support the conclusions?

Reviewer #1: Yes

Reviewer #2: Yes

2. Has the statistical analysis been performed appropriately and rigorously? 

Reviewer #1: Yes

Reviewer #2: Yes

3. Have the authors made all data underlying the findings in their manuscript fully available?

Reviewer #1: Yes

Reviewer #2: Yes

4. Is the manuscript presented in an intelligible fashion and written in standard English?

Reviewer #1: No

Reviewer #2: Yes

5. Review Comments to the Author

Reviewer #1: Comments to the Author

Proesmans et al., report relevant data on risk factors for hospitalization and use of medical resources in children <5 years old in Belgium. The study nevertheless requires reorganization, better consistency and faithful presentation of the dependent and independent variables in the methodology and results sections.

Methodology

L97: In general, it is preferred to describe the following information: (company name, City, State. Country) for company information of the experimental kit used.

L105-106: what was your definition of LRTI?

L99: Is a 2–7-day follow-up long enough to expect to observe factors associated with the length of hospital stay and the use of medical resources?

Authors should clearly group together the potential risk factors for hospitalization and use of medical resources due to HRSV. That is to say socio-demographic factors, signs and symptoms, comorbidities, duration of symptoms before hospitalization, severity (PES, PES3).

The authors must clarify the dependent variables, i.e. hospitalization (length of hospitalization) and use of medical resources (mechanical ventilation, oxygen supplementation, duration of oxygen supplementation, reception of supplementation oxygen on day 1, concomitant medication, doctor's visit for ARI before hospitalization).

L140-143: this part should be presented in data analysis or socio-demographic data.

The authors should explain the orientation of the choice of the Wilcoxon test, chi square test, log rank and logistic regression.

Reviewer #2: Comments to Authors:

The study demonstrates the issue well in disease severity and medical resource utilization of hospitalized RSV-positive children. Overall, this is a well-written manuscript and contributes valuable respiratory data in pediatric clinical practice. The methods are generally appropriate, although authors should clarify a few details and provide a rationale for using analytical methods to measure medical resource utilization parameters.

Introduction

p.9, paragraph 3: Since the authors chose the age as a parameter for medical resource utilization categorized into 0-<3 months, 3-6 months, 6-<12 months, 12-<48 months, readers would want to see background epidemiological data of RSV in these age groups of children.

Methods

Study design

p.11, lines 93-95: Please explain on what grounds these hospitals were selected. Are these hospitals located in RSV endemic areas in the country?

p.11, line 96: Since authors addressed epidemic seasons in the Introduction of the study, the S1 figure is unnecessary. Instead, please add study start and end dates for each respective year.

Study population

p.12, line 105: Please give more specifics of study subjects. Are all pediatric patients included who met inclusion criteria? Readers would want to know why the recruited subjects are only 75 when the RSV detection rate is almost 80% of children under five with severe acute respiratory infection in Belgium generally. (Subissi, L., Bossuyt, N., Reynders, M., Gérard, M., Dauby, N., Bourgeois, M., ... & Barbezange, C. (2020). Capturing respiratory syncytial virus season in Belgium using the influenza severe acute respiratory infection surveillance network, season 2018/19. Eurosurveillance, 25(39), 1900627.)

Demographics and clinical characteristics

p.12, line 118: Did authors consider including re-infection of RSV in underlying conditions?

Medical resource utilization

p.13, line 138: As indications of medical resource utilization might differ depending on the country or context, please specify indications for supplementary oxygen provided for patients involved in the study. Also, authors might need to consider this indication as one of the covariates in the analysis if the practice differs across hospitals. Supplementary oxygen is a highly vulnerable treatment depending on the capacity of the hospitals, especially in developing countries.

Statistical analysis

p.13, line 147 and line 152: Please briefly explain why Kaplan-Meier survival analysis and Cox proportional hazard regression models were performed with distinction. Please indicate all the variables included in each analysis respectively.

p. 14, line 163: Please explain why p-values without multiple comparison corrections were reported.

Results

p. 15, line 170: Readers would want to know why children above 41 months did not take part in the study as inclusion criteria were pediatric patients up to 5 years old.

Table 1

Please clarify breastfeeding? Does it indicate breastfeeding at the point of the data collection?

Clinical disease kinetics

p.19, line 231-233: As shown in Fig 2, the ability to feed and wheezing were mentioned, but how about rales, rhonchi, or others as it appears to be presented more in patients with underlying risk factors too?

Discussion

p.23, paragraph 4: In case authors do not have data on bacterial co-infection, it might look misguiding to state that 41.3% of the patients prescribed with antibiotics highlight overuse of antibiotics. If the authors have more explanation in this regard, please do so. Otherwise, please revise.

p.24, line 330: Please discuss more on the limitation of the study, including potential sources of bias.

Minor comments:

Please correct typos such as:

p. 7, line 45: “PSE3” to “PES3”

p.8, line 59: “score” to “score.”

p.26, line 362: “children” to “children.”

6. PLOS authors have the option to publish the peer review history of their article (what does this mean?). If published, this will include your full peer review and any attached files.

Reviewer #1: No

Reviewer #2: **Yes: **Amarjargal Dagvadorj

---

## [Author Response · Author response to Decision Letter 0]

11 Jan 2022

Reviewer 1: 

We appreciate your thoughtful review of the manuscript and the responses to your comments are provided below

Comment 1: Proesmans et al., report relevant data on risk factors for hospitalization and use of medical resources in children <5 years old in Belgium. The study nevertheless requires reorganization, better consistency, and faithful presentation of the dependent and independent variables in the methodology and results sections.

Response: Based on your comments, we have made the necessary changes and hope the manuscript meets your expectations

Comment 2: Methodology - L97: In general, it is preferred to describe the following information: (company name, City, State. Country) for company information of the experimental kit used.

Response: Either the standard of care quantitative RT-PCR (qRT-PCR) test which concerns a semi-quantitative ISO-accredited LDT (laboratory-developed test)., or a study specific Sofia RSV fluorescent immunoassay (SOFIA®RSV tests, Quidel) were used. The data in the manuscript has been modified to include the details

Comment 3: L105-106: what was your definition of LRTI?

Response: No protocol specific definition of LRTI was provided, the diagnosis of LRTI was done based on clinical decision. In general practice, LRTI clinical diagnosis is based on the presence of symptoms such as: respiratory distress (tachypnea or retraction and/or abnormalities on auscultation).

Comment 4: L99: Is a 2–7-day follow-up long enough to expect to observe factors associated with the length of hospital stay and the use of medical resources?

Response: Yes, we are convinced that in this study the follow-up period of 7 days is sufficient. 

(1) All independent variables used in the modelling, on both length of hospital stay and the use of supplemental oxygen, use factors observed at baseline (day 1 in the hospital) namely demographic variables [age, sex], comorbidities [underlying risk], duration of symptoms before hospitalization, signs, and symptoms severity [individual and total PES3] and are thus not influenced by the length of study. 

(2) We agree that stopping the follow-up period at day 7 could imply that the actual length of stay is not observed for patients who would be longer hospitalized. In this study, only 8/74 (10.8 %) subjects had a length of stay of longer than 7 days, and thus had a censored length of stay. The applied Cox proportional hazards model accounts for these censored observations. 

For modelling of the length of hospital stay, results of the Cox proportional hazards model are now included in the Supplementary Materials Table S8. Results are qualitative similar as results of the logistic regression. This table has been mentioned in the section on ‘Factors affecting length of hospitalization’ in the manuscript.

Comment 5: Authors should clearly group together the potential risk factors for hospitalization and use of medical resources due to HRSV. That is to say socio-demographic factors, signs and symptoms, comorbidities, duration of symptoms before hospitalization, severity (PES, PES3).

Response: We agree with the reviewer. The text in the Statistical Analysis section (Page no 14/ line no 161) has changed to: “The following, at baseline available, independent variables (covariates) were considered: demographic variables [age, sex], comorbidities [underlying risk], duration of symptoms before hospitalization, signs and symptoms severity [individual PES3 items and total-PES3 score on day 1]. In addition, reception of oxygen supplementation on day 1 (Yes/No) was included as a covariate for the modelling of LOH. Three options were explored to incorporate the PES score on day 1: (1) the total PES score by summing the 8 individual components; (2) scores on the individual components of ability to feed, dyspnea and respiratory effort; and (3) the PES3 score. The results of options (2) and (3) are presented here.”

Comment 6: The authors must clarify the dependent variables, i.e. hospitalization (length of hospitalization) and use of medical resources (mechanical ventilation, oxygen supplementation, duration of oxygen supplementation, reception of supplementation oxygen on day 1, concomitant medication, doctor's visit for ARI before hospitalization).

Response: We agree with the reviewer. The text in the Statistical Analysis section (Page 14, line no 155) has been changed to: “Univariate and multivariate logistic regression models were applied to investigate whether covariates impact the dependent variables LOH (categorized as LOH ≤4 days and LOH >4 days) and the need for supplemental oxygen (subject received supplemental oxygen during hospitalization: Yes/No). Univariate and multivariate Cox proportional hazard regression models were implemented on the LOH (in days) and length of oxygen supplementation (in days).”

Comment 7: L140-143: this part should be presented in data analysis or socio-demographic data.

Response: Thanks for the suggestion, the text has now been moved to the Statistical analysis section

Comment 8: The authors should explain the orientation of the choice of the Wilcoxon test, chi square test, log rank and logistic regression.

Response: As mentioned in Statistical analysis section, all significance tests were 2-sided with a 5% significance level, where applicable. The ‘where applicable’ refers to the fact that chi-square test for independence of two categorical variables is always one-sided.

 

Reviewer 2:

The study demonstrates the issue well in disease severity and medical resource utilization of hospitalized RSV-positive children. Overall, this is a well-written manuscript and contributes valuable respiratory data in pediatric clinical practice. The methods are generally appropriate, although authors should clarify a few details and provide a rationale for using analytical methods to measure medical resource utilization parameters.

Comment 1:

Introduction

p.9, paragraph 3: Since the authors chose the age as a parameter for medical resource utilization categorized into 0-<3 months, 3-6 months, 6-<12 months, 12-<48 months, readers would want to see background epidemiological data of RSV in these age groups of children.

Response: The following text along with the corresponding references has been added in the introduction (Page 9, line no 69):

It has been documented that the clinical presentation of RSV infection in children differs according to age and may be influenced by the differences in their immune reaction to RSV. In a metareview, (Bont L et al., Defining the Epidemiology and Burden of Severe Respiratory Syncytial Virus Infection Among Infants and Children in Western Countries. Infect Dis Ther. 2016 Sep;5(3):271-98.) it was shown that the annual RSV hospitalization rates decreased with increasing age and varied by a factor of 2-3. Risks factors associated with RSV related medical resources utilization included: male sex; age <6 months; birth during the first half of the RSV season; crowding/siblings; and day-care exposure (high strength of evidence).

Comment 2: Methods

Study design

p.11, lines 93-95: Please explain on what grounds these hospitals were selected. Are these hospitals located in RSV endemic areas in the country?

Response: The study had a central site that coordinated the additional sites included. The hospitals were in Flanders, and through their locations allowed a good geographical coverage of the seasonal RSV circulation in that region. The study sites included tertiary academic center/university, large regional and small regional hospitals.

Comment 3: p.11, line 96: Since authors addressed epidemic seasons in the Introduction of the study, the S1 figure is unnecessary. Instead, please add study start and end dates for each respective year.

Response: We have added the study start and end date. The text now reads as:

This exploratory, prospective, multicenter study (NCT02133092) enrolled patients from clinical pediatric wards of four Belgian hospitals (one tertiary academic center and three regional hospitals: UZ Leuven, AZ Sint-Jan Brugge – Oostende campus Brugge, AZ Sint-Jan Brugge - Oostende campus Henri Serruys, and Heilig Hart Ziekenhuis Lier) during the 2013–2014 and 2014–2015 RSV epidemic seasons (Study initiated: 17 December 2013 and Study completed: 21 January 2015)

Comment 4: Study population

p.12, line 105: Please give more specifics of study subjects. Are all pediatric patients included who met inclusion criteria? Readers would want to know why the recruited subjects are only 75 when the RSV detection rate is almost 80% of children under five with severe acute respiratory infection in Belgium generally. (Subissi L, et al. Capturing respiratory syncytial virus season in Belgium using the influenza severe acute respiratory infection surveillance network, season 2018/19. Euro Surveill. 2020 Oct; 25(39): 1900627.)

Response: All pediatric patients included in the study met the inclusion criteria, however, not all patients that met the inclusion criteria were enrolled in the study. Furthermore, the study was not designed to measure RSV positivity rate in all patients hospitalized. The patients were not enrolled during the Weekends and public holidays. 

One possible explanation for the difference between the positivity rate found in our research and the Subissi et al 2020 reference, could be different case definitions used for eligibility across the two studies, with any ARI for our study and SARI case definition for the referenced study.

We would like to highlight that in Belgium, similar to other countries, the positivity rate in surveillance data, are between 20-50%, depending on the intensity of RSV circulation, and aligned with our findings. (https://www.tandfonline.com/doi/epub/10.1080/17843286.2018.1492509?needAccess=true)

Comment 5: Demographics and clinical characteristics

p.12, line 118: Did authors consider including re-infection of RSV in underlying conditions?

Response: We have no information on whether the RSV event for which the child was included in the study is a possible re-infection of RSV. Indeed, this would have been valuable information, however, re-infections or prior hospitalization due to RSV were not captured.

Comment 6: Medical resource utilization

p.13, line 138: As indications of medical resource utilization might differ depending on the country or context, please specify indications for supplementary oxygen provided for patients involved in the study. Also, authors might need to consider this indication as one of the covariates in the analysis if the practice differs across hospitals. Supplementary oxygen is a highly vulnerable treatment depending on the capacity of the hospitals, especially in developing countries.

Response: The decision to provide O2 supplementation was done according to the Standard of care. Whereas there was no harmonization on O2 supplementation start and stop decisions, in general, considering that the participating sites were located in the same region /country, they likely had similar local guidelines for O2 supplementation, with 92% SpO2 used as a cut-off to guide hospitalization and request for O2 supplementation.

In this study we had 4 sites with, respectively, 14, 10, 24 and 27 subjects. Proportions of subjects receiving oxygen supplementation differs, markedly, by site with respectively 92.9% (13/14), 20.0% (2/10), 41.7% (10/24) and 70.4% (19/27). Further, it was observed that the site with highest proportions of oxygen supplementation also had the highest proportion of youngest subjects. However, as can be observed from the table below, overall, there is a trend in all sites that mainly the younger patients received supplemental oxygen.

 Received oxygen supplementation

 All Age,

0 - <3 months Age,

3 - <6 months Age,

6 - <12 months Age,

12 - <48 months

Site 1 (n=14) 13 (92.9 %) 9 / 9 (100 %) 1 / 2 (50.0 %) - 3 / 3 (100 %)

Site 2 (n=10) 2 (20.0 %) 1 / 2 (50.0 %) 0 / 1 (0.0 %) 0 / 2 (0.0 %) 1 / 5 (20.0 %)

Site 3 (n=24) 10 (41.7 %) 4 / 6 (66.7 %) 3 / 6 (50.0 %) 1 / 4 (25.0 %) 2 / 8 (25.0 %)

Site 4 (n=27) 19 (70.4 %) 9 / 11 (81.2 %) 3 / 6 (50.0 %) 6 / 7 (85.7 %) 1 / 2 (50.0 %)

Based on clinical input that an association exists between age and receiving supplemental oxygen and the results in the table above, it was decided not to include site as a covariate in the statistical models. Including site as a covariate could possibly dilute the effect of age. In addition, the manner the different sites recruited participants could be different (some sites could have included mainly sicker patients that have a higher probability of receiving supplemental oxygen). This difference is a potential factor for bias when site would be included as a covariate in the modelling. 

This was also added to the Discussion section of the manuscript (see also below on the comment on potential sources of bias raised by the reviewer).

Comment 7: Statistical analysis

p.13, line 147 and line 152: Please briefly explain why Kaplan-Meier survival analysis and Cox proportional hazard regression models were performed with distinction. Please indicate all the variables included in each analysis respectively.

Response: Kaplan-Meier analysis can only be used for univariate analysis (thus one covariate at a time). Whereas the Cox proportional hazard regression model can incorporate multiple variables at the same moment.

We adjusted the Statistical Analysis section such that it is more clearly presented which variables are included in each analysis: “Univariate and multivariate logistic regression models were applied to investigate whether covariates impact the dependent variables LOH (categorized as LOH ≤4 days and LOH >4 days) and the need for supplemental oxygen (subject received supplemental oxygen during hospitalization: Yes/No). Univariate and multivariate Cox proportional hazard regression models were implemented on the LOH (in days) and length of oxygen supplementation (in days).

The following, at baseline available, independent variables (covariates) were considered: demographic variables [age, sex], comorbidities [underlying risk], duration of symptoms before hospitalization, signs and symptoms severity [individual PES3 items and total-PES3 score on day 1]. In addition, reception of oxygen supplementation on day 1 (Yes/No) was included as a covariate for the modelling of LOH. Three options were explored to incorporate the PES score on day 1: (1) the total PES score by summing the 8 individual components; (2) scores on the individual components of ability to feed, dyspnea and respiratory effort; and (3) the PES3 score. The results of options (2) and (3) are presented here.”

Comment 8: p. 14, line 163: Please explain why p-values without multiple comparison corrections were reported.

Response: It is common practice that multiple testing correction is not applied to multiple linear regression. We want to point to the following publication on multiple comparison [‘Adjust for Multiple Comparisons? It’s Not That Simple’ by A. Althouse, doi: https://doi.org/10. 1016/j.athoracsur.2015.11.024]. In our and the author’s opinion, the best approach is simply to (1) describe what was done in a study; (2) report point estimates, confidence intervals, and p-values; and (3) let readers use their own judgment about the relative weight of the conclusions. The author and others argue that because adjustment for multiple comparisons has several practical considerations that make the concept unreasonable to apply to every research paper, and particularly not to exploratory studies. 

In confirmatory studies, which may lead to a change in clinical practice or approval of a new treatment, it is more important to guard against the possibility of false-positive results. When it comes to exploratory studies or post-hoc analysis of existing data, though, a strict adjustment for multiple comparisons is less critical, as long as the manuscript contains a clear statement acknowledging that.

Comment 9: Results: p. 15, line 170: Readers would want to know why children above 41 months did not take part in the study as inclusion criteria were pediatric patients up to 5 years old.

Response: Although the inclusion criteria were open to 5-year-old, most of the patients hospitalized were <1 year old; 41 months was the maximum observed age in the study. This is in alignment with what has been published in the field, with RSV burden of hospitalization been reported mainly in the very young children. 

Comment 11: Table 1: Please clarify breastfeeding. Does it indicate breastfeeding at the point of the data collection?

Response: Yes, this was breastfeeding status at enrollment

Comment 12: Clinical disease kinetics

p.19, line 231-233: As shown in Fig 2, the ability to feed and wheezing were mentioned, but how about rales, rhonchi, or others as it appears to be presented more in patients with underlying risk factors too?

Response: Thank you for the suggestion, changes in rales and rochis and others have now been highlighted as well. Corresponding change in the text is as below

Higher scores i.e., worse condition in the ability to feed item for the 3–6 months group, wheezing; changes in rales and rochi, dyspnea, respiratory effort, and otitis for patients with underlying risk factors show that certain functions were affected more in certain groups.

Comment 13: Discussion

p.23, paragraph 4: In case authors do not have data on bacterial co-infection, it might look misguiding to state that 41.3% of the patients prescribed with antibiotics highlight overuse of antibiotics. If the authors have more explanation in this regard, please do so. Otherwise, please revise.

Response: - No data on bacterial co-infection was available. However, we based our statement on the following considerations:

• The frequency of RSV and bacterial co-infection in children is relatively low. Numerous studies have shown that the occurrence of a secondary or concurrent bacterial infection in hospitalized children with RSV lower respiratory tract disease is <1% (https://adc.bmj.com/content/89/4/363?ijkey=04c663ca8ef80c49237de729d2677f9842cca96f&keytype2=tf_ipsecsha)

• The over-use of antibiotics for treatment of RSV disease was highly documented in the field.

- Reducing Antibiotic Use in Respiratory Syncytial Virus-A Quality Improvement Approach to Antimicrobial Stewardship - PubMed (nih.gov)

- Antibiotic Overuse in Children with Respiratory Syncytial Virus Lower Respiratory Tract Infection - PubMed (nih.gov)

However, the following change in text (Page 23, line no 320) is made, to acknowledge the absence of bacterial co-infection data: 

We did not collect data on bacterial co-infection; however, we suspect overuse of antibiotics based on the facts that numerous studies have shown that the occurrence of a secondary or concurrent bacterial infection in hospitalized children with RSV lower respiratory tract disease Nevertheless, frequency of RSV and bacterial co-infection in children is very low. The overuse of antibiotics for treatment of RSV disease has been documented before.

Comment 14: p.24, line 330: Please discuss more on the limitation of the study, including potential sources of bias.

Response: We have modified the limitations to identify the potential sources of bias. The limitations paragraph now reads as below:

Due to practical reasons, only a subset of patients admitted for RSV infections were enrolled in the study, therefore we cannot comment on the generalizability of the results. In addition, the way different sites recruited participants could be different (some sites could have included mainly sicker patients that have a higher probability of receiving supplemental oxygen). We realized that the patients included may not be representative for all RSV related hospitalizations in the participating sites. Since a strict protocol on initiation and discontinuation of oxygen supplementation was not implemented, all participating hospitals followed their in-house protocol, which may vary from each other. The participating hospitals also did not have documented discharge criteria. This difference is a potential factor for bias introduced by site effects. Although not excluded in the protocol, no intensive care cases were included, hence most severe cases were not captured.

Comment 15: Minor comments:

Please correct typos such as:

p. 7, line 45: “PSE3” to “PES3” 

p.8, line 59: “score” to “score.”

p.26, line 362: “children” to “children.”

Response: Thanks for noting. We have now corrected the typos.

---

## [Decision Letter · Decision Letter 1]

25 Feb 2022

PONE-D-21-25349R1Risk factors for disease severity and increased medical resource utilization in respiratory syncytial virus (+) hospitalized children: a descriptive study conducted in four Belgian hospitalsPLOS ONE

Dear Dr. Ispas,

Thank you for submitting your manuscript to PLOS ONE. After careful consideration, we feel that it has merit but does not fully meet PLOS ONE’s publication criteria as it currently stands. Therefore, we invite you to submit a revised version of the manuscript that addresses the points raised during the review process. Please submit your revised manuscript by Apr 11 2022 11:59PM. If you will need more time than this to complete your revisions, please reply to this message or contact the journal office at plosone@plos.org. Please include the following items when submitting your revised manuscript:A rebuttal letter that responds to each point raised by the academic editor and reviewer(s). You should upload this letter as a separate file labeled 'Response to Reviewers'.A marked-up copy of your manuscript that highlights changes made to the original version. You should upload this as a separate file labeled 'Revised Manuscript with Track Changes'.An unmarked version of your revised paper without tracked changes. You should upload this as a separate file labeled 'Manuscript'.If applicable, we recommend that you deposit your laboratory protocols in protocols.io to enhance the reproducibility of your results. Protocols.io assigns your protocol its own identifier (DOI) so that it can be cited independently in the future. For instructions see: https://journals.plos.org/plosone/s/submission-guidelines#loc-laboratory-protocols. Additionally, PLOS ONE offers an option for publishing peer-reviewed Lab Protocol articles, which describe protocols hosted on protocols.io. Read more information on sharing protocols at https://plos.org/protocols?utm_medium=editorial-email&utm_source=authorletters&utm_campaign=protocols.

We look forward to receiving your revised manuscript.

Kind regards,

Brenda M. Morrow, PhD

Academic Editor

PLOS ONE

Journal Requirements:

Reviewers' comments:

Reviewer's Responses to Questions

**Comments to the Author**

1. If the authors have adequately addressed your comments raised in a previous round of review and you feel that this manuscript is now acceptable for publication, you may indicate that here to bypass the “Comments to the Author” section, enter your conflict of interest statement in the “Confidential to Editor” section, and submit your "Accept" recommendation.

Reviewer #1: (No Response)

Reviewer #2: All comments have been addressed

2. Is the manuscript technically sound, and do the data support the conclusions?

Reviewer #1: Yes

Reviewer #2: Yes

3. Has the statistical analysis been performed appropriately and rigorously? 

Reviewer #1: Yes

Reviewer #2: Yes

4. Have the authors made all data underlying the findings in their manuscript fully available?

Reviewer #1: Yes

Reviewer #2: Yes

5. Is the manuscript presented in an intelligible fashion and written in standard English?

Reviewer #1: Yes

Reviewer #2: Yes

6. Review Comments to the Author

Reviewer #1: Comments to the Author

Thanks to the authors for responding to the questions raised. There are, however, the questions below that I would like further clarification.

Methodology

L97: In general, it is preferred to describe the following information: (company name, City, State. Country) for company information of the experimental kit used.

Thank you for modifying the document in accordance with the comment above. However, it will be better to also specify in the main manuscript that the qRT-PCR assay was homemade.

L99: Is a 2–7-day follow-up long enough to expect to observe factors associated with the length of hospital stay and the use of medical resources?

Thank you for providing the explanations in the letter in relation to the comment above. You affirm that you are convinced that 7 days is long enough to observe all the outcomes of the use of health resources. It would be greatly appreciated if you provide a reference that supports your statement. Also, the explanation that 8/74 participants had more than 7 days in hospital would be important to mention in the main manuscript.

Reviewer #2: The authors have clarified all of the questions I raised in my previous review except one problem.

Also, the two copies of the manuscript provided were somewhat different. I would suggest using a file named “Observe001_MSS_Resubmission version_clean copy” instead of a file named “Observe001_MSS_PLOS ONE” as the latter does not seem to reflect all the revisions made.

In general, the paper appears to be worthwhile, and I would accept after addressing the following issue:

Table 1: Please add and revise the appropriate term for the “breastfeeding” variable. Although the authors stated that the variable shows breastfeeding status at enrollment, how about older children who had stopped breastfeeding as your study subjects are up to 41 months old? If you marked them as “no” in the breastfeeding variable, that would be a major concern. Showing the breastfeeding duration is vital as a sufficient breastfeeding period can provide a long-term protective effect against respiratory tract infections. Authors might also need to include breastfeeding as one of the independent variables in multivariate analysis. Because it is well reported that breastfeeding is significant in reducing the rate of severe RSV infection cases. ( please refer to https://www.thelancet.com/series/breastfeeding)

7. PLOS authors have the option to publish the peer review history of their article (what does this mean?). If published, this will include your full peer review and any attached files.

Reviewer #1: No

Reviewer #2: **Yes: **Amarjargal Dagvadorj, MD, MSc, DrPH

---

## [Author Response · Author response to Decision Letter 1]

31 Mar 2022

Reviewer #1: Comments to the Author

Thanks to the authors for responding to the questions raised. There are, however, the questions below that I would like further clarification.

Methodology

L97: In general, it is preferred to describe the following information: (company name, City, State. Country) for company information of the experimental kit used.

Thank you for modifying the document in accordance with the comment above. However, it will be better to also specify in the main manuscript that the qRT-PCR assay was homemade.

Response: Thank you for your suggestion, we have specified that qRT-PCR assay was homemade in the manuscript.

L99: Is a 2–7-day follow-up long enough to expect to observe factors associated with the length of hospital stay and the use of medical resources?

Thank you for providing the explanations in the letter in relation to the comment above. You affirm that you are convinced that 7 days is long enough to observe all the outcomes of the use of health resources. It would be greatly appreciated if you provide a reference that supports your statement. Also, the explanation that 8/74 participants had more than 7 days in hospital would be important to mention in the main manuscript.

Response: Thank you for your suggestion, we already have this statement in the result section of the manuscript (L225, hospitalization paragraph of result section). We have now added the additional explanation along with the references that would support the length of hospital stay we used in this study. 

1) The objective of the study was to describe clinical and medical burden during the hospitalization. However, most of the children are discharged with ongoing signs/symptoms that may drive additional medical resources utilization after hospital discharge. Due to the study design, we could not capture the MRUs after discharge, this being another limitation of the study. We have included this in the discussion section

2) We consider that 7 days follow-up is sufficient to monitor the use of hospital health resources in the enrolled population, besides only 8/74 participants enrolled were hospitalized for more than 7 days. We have also included some literature in the discussion section to support the length of the hospital stay observed in our study,[median of 5.0 days (2–7) d CI], which is similar to the length of hospitalization observed in a Belgium study (LOH: median 5 days, 3-11 CI) (Subissi et al., Euro Surveill. 2020 Oct 1; 25(39): 1900627) and in Spain (LOH: mean 4.8 +/- 11d) (Viguria et al., PLoS One. 2018 Nov 15;13(11):e0206474)

3) The logistic regression analysis on the probability of hospitalization length (LOH ≤ 4 days, LOH > 4 days) and probability of requiring oxygen supplementation, as well as the Cox regression models on the length of hospitalization and length of oxygen supplementation only have baseline covariates included. Thus, the included covariates are independent on the length of follow-up of the patients. The applied Cox proportional hazards model account for censored observations, and thus all available information of the 8 patients with more than 7 days of hospitalizations is included.

 

Reviewer #2: The authors have clarified all of the questions I raised in my previous review except one problem.

Also, the two copies of the manuscript provided were somewhat different. I would suggest using a file named “Observe001_MSS_Resubmission version_clean copy” instead of a file named “Observe001_MSS_PLOS ONE” as the latter does not seem to reflect all the revisions made.

In general, the paper appears to be worthwhile, and I would accept after addressing the following issue:

Response: Thank you for your suggestion and accepting the paper, we have renamed the file to, ‘Observe001_MSS_Resubmission version_clean copy’ as suggested. We have also verified and checked if the current version reflects all the revisions, and we are able to see the revisions.

Table 1: Please add and revise the appropriate term for the “breastfeeding” variable. Although the authors stated that the variable shows breastfeeding status at enrolment, how about older children who had stopped breastfeeding as your study subjects are up to 41 months old? If you marked them as “no” in the breastfeeding variable, that would be a major concern. Showing the breastfeeding duration is vital as a sufficient breastfeeding period can provide a long-term protective effect against respiratory tract infections. Authors might also need to include breastfeeding as one of the independent variables in multivariate analysis. Because it is well reported that breastfeeding is significant in reducing the rate of severe RSV infection cases. (Please refer to https://www.thelancet.com/series/breastfeeding)

Response: Thank you for suggestion and valuable input regarding the breastfeeding being significant in reducing the rate of severe RSV infection cases.

We have changed the term to ‘currently breastfed’ in the manuscript. Unfortunately, we don’t have information on breastfeeding status or length in the past. We added in the discussion that it is a limitation that we cannot study the possible long-term protective effect of breastfeeding on hospitalization characteristics.

The relevant literature published in this field assessed breastfeeding as a risk factor for progression to hospitalization. However, studies that assessed breastfeeding as a risk factor for severity of disease and length of hospitalization and MRU during hospitalization are limited. Below are some relevant references 

1) This study suggests that there was not a significant association between breastfeeding and bronchiolitis severity score or length of hospital stay (Vereen S, Gebretsadik T, Hartert TV et al., Pediatr Infect Dis J. 2014 Sep;33(9):986-8).

2) This article states breastfeeding (<2months or not) was a predictor of progression to hospitalization (Blanken MO, Koffijberg H, Nibbelke EE et al., PLoS One. 2013;8(3):e59161)

3) In a meta-analysis, breastfeeding was observed to be associated with RSV ALRI. However, the definition used for breastfeeding in the reviewed articles was different - No breastfeeding = [no breastfeeding for first 14 days, <3 months breastfeeding or lack of exclusive breastfeeding] (Shi T, Balsells E, Wastnedge E, Singleton R et al., J Glob Health. 2015 Dec;5(2):020416)

Current breastfeeding status (yes/no) is associated with age (see Table S3) with higher proportions of currently being breastfed in the youngest age group. 

A Kaplan-Meier analysis showed no differences between current breastfeeding status and length of hospital stay (p=0.55). We also included current breastfeeding status as one of the independent variables in both a univariate and the multivariate regression analyses for length of hospital stay. Current breastfeeding status showed no differences in these regression models.

In our study, patients with current breastfeeding have a longer length of oxygen supplementation (Kaplan-Meier analysis, p=0.046). This result, however, is confounded with age. A univariate logistic regression analysis for the probability of receiving oxygen supplementation shows a significant effect of breastfeeding status. Patients receiving currently breastfeeding have a higher probability of receiving oxygen with OR=3.33 [1.18 – 10.51]. However, in a multivariate logistic regression (and thus corrected for age) no effect is observed anymore of current breastfeeding status (OR = 0.73 [0.15 – 3.40]). Similarly, in the multivariate Cox regression model no significant effect of breastfeeding status is observed (HR = 1.19 [0.64 – 2.22]).

---

## [Decision Letter · Decision Letter 2]

3 May 2022

Risk factors for disease severity and increased medical resource utilization in respiratory syncytial virus (+) hospitalized children: a descriptive study conducted in four Belgian hospitals

PONE-D-21-25349R2

Dear Dr. Ispas,

We’re pleased to inform you that your manuscript has been judged scientifically suitable for publication and will be formally accepted for publication once it meets all outstanding technical requirements.

Kind regards,

Brenda M. Morrow, PhD

Academic Editor

PLOS ONE

Reviewers' comments:

Reviewer's Responses to Questions

**Comments to the Author**

1. If the authors have adequately addressed your comments raised in a previous round of review and you feel that this manuscript is now acceptable for publication, you may indicate that here to bypass the “Comments to the Author” section, enter your conflict of interest statement in the “Confidential to Editor” section, and submit your "Accept" recommendation.

Reviewer #1: All comments have been addressed

Reviewer #2: All comments have been addressed

2. Is the manuscript technically sound, and do the data support the conclusions?

Reviewer #1: Yes

Reviewer #2: Yes

3. Has the statistical analysis been performed appropriately and rigorously? 

Reviewer #1: Yes

Reviewer #2: Yes

4. Have the authors made all data underlying the findings in their manuscript fully available?

Reviewer #1: Yes

Reviewer #2: Yes

5. Is the manuscript presented in an intelligible fashion and written in standard English?

Reviewer #1: Yes

Reviewer #2: Yes

6. Review Comments to the Author

Reviewer #1: I enjoyed reviewing this relevant manuscript on risk factors for hospitalization and use of medical resources in children. Thank you for the invitation to review. Regards.

Reviewer #2: The authors revised the article well as per my suggestions. This work contributes towards further improvement of treatment for young children with RSV infection.

7. PLOS authors have the option to publish the peer review history of their article (what does this mean?). If published, this will include your full peer review and any attached files.

Reviewer #1: No

Reviewer #2: **Yes: **Amarjargal Dagvadorj

---

## [Editor Report · Acceptance letter]

26 May 2022

PONE-D-21-25349R2 

Risk factors for disease severity and increased medical resource utilization in respiratory syncytial virus (+) hospitalized children: a descriptive study conducted in four Belgian hospitals

Dear Dr. Ispas:

I'm pleased to inform you that your manuscript has been deemed suitable for publication in PLOS ONE. Congratulations! Your manuscript is now with our production department. 

Kind regards, 

on behalf of

Professor Brenda M. Morrow 

Academic Editor

PLOS ONE